# Achimota Pararubulavirus 3: A New Bat-Derived Paramyxovirus of the Genus *Pararubulavirus*

**DOI:** 10.3390/v12111236

**Published:** 2020-10-30

**Authors:** Kate S. Baker, Mary Tachedjian, Jennifer Barr, Glenn A. Marsh, Shawn Todd, Gary Crameri, Sandra Crameri, Ina Smith, Clare E.G. Holmes, Richard Suu-Ire, Andres Fernandez-Loras, Andrew A. Cunningham, James L.N. Wood, Lin-Fa Wang

**Affiliations:** 1Department of Veterinary Medicine, University of Cambridge, Madingley Rd, Cambridge CB3 0ES, UK; jlnw2@cam.ac.uk; 2Institute of Zoology, Zoological Society of London, London NW1 4RY, UK; aferlasvet@hotmail.com (A.F.-L.); a.cunningham@ioz.ac.uk (A.A.C.); 3Institute for Infection, Veterinary and Ecological Sciences, University of Liverpool, Liverpool L69 7ZB, UK; 4CSIRO Health and Biosecurity, Australian Animal Health Laboratory, Portarlington Road, East Geelong, VIC 3220, Australia; Mary.Tachedjian@csiro.au (M.T.); Jennifer.Barr@csiro.au (J.B.); Glenn.Marsh@csiro.au (G.A.M.); Shawn.Todd@csiro.au (S.T.); gary.Crameri@csiro.au (G.C.); Sandra.Crameri@csiro.au (S.C.); clare.holmes@csiro.au (C.E.G.H.); 5CSIRO Health & Biosecurity, Clunies Ross Street, Black Mountain, ACT 2601, Australia; Ina.Smith@csiro.au; 6Wildlife Division of the Forestry Commission, Accra PO Box M239, Ghana; suuire@hotmail.com; 7Noguchi Memorial Institute for Medical Research, University of Ghana, Legon, Accra PO Box LG 581, Ghana; 8Programme in Emerging Infectious Diseases, Duke-NUS Graduate Medical School, Singapore 169857, Singapore

**Keywords:** virus, zoonosis, bat, pararubulavirus, paramyxovirus, molecular detection, virus discovery, electron microscopy, genomics, primary cell lines

## Abstract

Bats are an important source of viral zoonoses, including paramyxoviruses. The paramyxoviral *Pararubulavirus* genus contains viruses mostly derived from bats that are common, diverse, distributed throughout the Old World, and known to be zoonotic. Here, we describe a new member of the genus Achimota pararubulavirus 3 (AchPV3) and its isolation from the urine of African straw-coloured fruit bats on primary bat kidneys cells. We sequenced and analysed the genome of AchPV3 relative to other *Paramyxoviridae*, revealing it to be similar to known pararubulaviruses. Phylogenetic analysis of AchPV3 revealed the failure of molecular detection in the urine sample from which AchPV3 was derived and an attachment protein most closely related with AchPV2—a pararubulavirus known to cause cross-species transmission. Together these findings add to the picture of pararubulaviruses, their sources, and variable zoonotic potential, which is key to our understanding of host restriction and spillover of bat-derived paramyxoviruses. AchPV3 represents a novel candidate zoonosis and an important tool for further study.

## 1. Introduction

Bats are an increasingly recognised source of zoonotic viruses that cause devastating disease outbreaks worldwide, including the recent Ebola virus outbreaks, Middle Eastern Respiratory Syndrome, and the ongoing COVID-19 epidemic, as well as many viruses of unknown zoonotic potential [1,2,3,4,5]. To better prepare for these outbreaks, it is helpful to know what viruses exist in bats and focus our efforts on those viral taxa that are known to cause infection in humans and other animals. In addition to coronaviruses and filoviruses, bats host a diverse range of paramyxoviruses, harbouring a greater diversity of this viral family than other mammalian orders [6]. Two paramyxoviruses, Hendra virus (HeV) and Nipah virus (NiV), infect a diverse range of mammalian hosts, causing significant and ongoing disease in humans and domestic animal species [7]. Thus, it is important that other potentially zoonotic paramyxoviruses from bats are studied.

Within the paramyxoviral subfamily, *Rubulavirinae*, the genus *Pararubulavirus* is comprised of viruses almost exclusively derived from bats, many of which are known to cause, or are likely capable of, zoonotic infection. Specifically, the first known pararubulavirus, Menangle pararubulavirus (MenPV), caused cross-species infections in both pigs and humans in Australia [8,9]. Subsequently, Tioman pararubulavirus (TioPV) was isolated from bats in Malaysia and serum neutralising antibodies were detected in humans living near the isolation site [10]. Achimota pararubulaviruses 1 and 2 (AchPV1 and AchPV2) were recently isolated from urine collected underneath bat roosts in Ghana, Africa, and the existence of neutralising antibodies in humans and morbid cross-species infections of laboratory animals suggests zoonotic infection with Achimota viruses is possible [11,12]. The Teviot virus was also recently isolated from urine collected underneath bat roosts in Australia (although no clinical disease accompanied seroconversion in laboratory animal infection studies) [13]. The only pararubulavirus that was not originally isolated from bats is the Sosuga virus (SosV), which caused febrile systemic illness in a wildlife biologist after handling a wide variety of wildlife species. This patient history and the phylogenetic relationship of SosV with bat-derived pararubulaviruses suggested the infection had been contracted from bats, which was later confirmed by molecular detection of the virus in samples from *Rousettus aegyptiacus* bats [14], adding to the weight of evidence that *Pararubulavirus* is a bat-derived paramyxoviral genus, the members of which are capable of zoonotic infection.

In addition to these isolated viruses, molecular studies suggest that a great deal more diversity of the *Pararubulavirus* genus exists in bats. Specifically, the full genome sequences of three Tuhoko pararubulaviruses (ThkPVs), derived from samples collected in China, along with partial nucleocapsid and polymerase gene sequencing, demonstrates the existence of several-fold more pararubulaviruses in various bat species across Asia, Africa, and Europe [6,15,16,17,18,19,20,21]. Collectively, these findings demonstrate pararubulaviruses to be widely distributed bat-derived viruses with known potential to cause zoonotic disease, but for which live isolates represent only a fraction of the known genetic diversity.

Here, we report the isolation of a novel *Pararubulavirus*, Achimota pararubulavirus 3 (AchPV3), from a large urban roost of African straw-coloured fruit bats (*Eidolon helvum*) in Ghana, Africa. We characterised the genome of AchPV3, determined its evolutionary relationships among the *Paramyxoviridae*, including for the receptor binding protein, and contextualized its isolation among the detection of contemporary, un-isolated paramyxoviruses detected through consensus PCR. AchPV3 represents a novel candidate zoonotic virus and our work here lays the important foundation for future more in-depth investigations of this, and other, newly discovered bat paramyxoviruses.

## 2. Materials and Methods

### 2.1. Cell Culture Conditions

Experiments described used either Vero cells (ATCC CCL-81), Vero-E6 cells (ATCC CRL-1586), or *Pteropus alecto* primary kidney (PaKi) cells [22] and were conducted under BSL3 conditions. Cells were grown in Dulbecco’s modified Eagle’s medium supplemented with F12-Ham (Sigma-Aldrich, Macquiarie Park, Australia), 10% foetal calf serum, double-strength antibiotic/antimycotic (200 U/mL penicillin, 200 µg/mL streptomycin, and 0.5 µg/mL fungizone amphotericin B; Gibco), and ciprofloxacin (10 µg/mL; MP Bio- medicals), at 37 °C in 5% CO_2_.

### 2.2. Urine Samples

Briefly, 72 pooled urine samples were collected from sheets underneath a colony of *E. helvum* in Accra, Ghana [23], as previously described [11,15].

### 2.3. Isolation Methods

Three passages of virus isolation were attempted on urine samples U34–U72 on PaKi monolayers, in the same manner and in parallel with attempts previously described on Vero cells [11]. The only discrepancy was that, for some samples (U38, 42, 48–49, 58–62, 70, 72), the flasks were frozen at −80 °C and thawed at room temperature prior to passage (to synchronise experiments). Supernatants of cultures showing signs of cytopathic effect (CPE) were tested for the presence of *Paramyxovirinae* RNA using the previously described RT-PCR [24]; the primers are in Table 1. PCR products were cloned (pGEM-T Easy, Promega, Madison, WI, USA) and capillary-sequenced for phylogenetic analysis, as previously described [15].

### 2.4. Isolate Propagation for Genome Sequencing and Electron Microscopy

Following the confirmation of a paramyxoviral isolate, the cell monolayer was scraped into the media and the material was frozen at −80 °C. After thawing at room temperature, 50 µL of this stock virus was added to 75 cm^2^ near-confluent monolayers of Vero E6 or PaKi cells in minimal media for 1 h with gentle rocking at 37 °C. Following the incubation, further media was added, and the cells were observed for cytopathic effect (CPE) daily.

### 2.5. Genomic Sequencing and Bioinformatic Analysis

Viral supernatant was harvested from PaKi cells 6 days post inoculation and Vero-E6 cells 13 days post inoculation. Whole-genome sequencing was performed as previously described [25], except total RNA was extracted from a 100 uL culture supernatant with Zymo’s Direct-zol RNA Mini kit without DNaseI digestion (Zymo Research, Irvine, CA, USA). Nextera XT DNA libraries (Illumina, San Diego, CA, USA) were sequenced on an Illumina MiniSeq Sequencing System and Mid Output Kit (300-cycles) generating 150 bp paired-end (PE) reads (done at CSIRO AAHL). Complete genome sequences were obtained with a previously established de novo assembly pipeline [26], except host subtraction was omitted. The de novo-assembled contigs were verified by mapping back the trimmed reads using the default settings of the CLC Genomics Workbench version 10.1.1 “Map Reads to Reference” tool in addition to 5′ and 3′ genome end determination and genome annotation. Predicted open reading frames were verified by querying the NCBI Nucleotide BLAST non-redundant database. The P gene RNA editing site was verified with the CLC Genomics Workbench v10.1.1 using the “Low Frequency Variant Detection” algorithm with a minimum frequency percentage of 0.01%. Other bioinformatic analyses of the genome sequences, including genome annotation and phylogenetic and protein sequence analyses, were performed as previously described [11].

### 2.6. Electron Microscopy

PaKi cells infected with AchPV3 and overlying media were examined by electron microscopy (EM). The cells were scrapped and pelleted using centrifugation (2000× *g* for 5 min) and the supernatant reserved for negative contrast EM. The pellet was processed into resin blocks for ultra-thin sectioning. After primary fixation in 2.5% glutaraldehyde and washing in a buffer, the pellet was post fixed in 1% osmium tetroxide. During these steps, 0.1 M Sorenson’s phosphate buffer (300 mOsm/kg, pH 7.2) was used. Following fixation, the pellet was dehydrated through an ethanol series of 70 to 100%, infiltrated, embedded, and polymerised in Spurr’s resin as per the manufacturer’s instructions (ProSciTech, product code:C035). Sections were stained using uranyl acetate followed by lead citrate and examined on a Philips CM120 transmission EM (FEI) at 120 kV. The reserved supernatant was absorbed into grids (ProSciTech, product code: GSCU400C-50) for 5 min and negative stained with Nano–W (Nanoprobes, item number: 2018-5 mL) for 1 min. The grids were examined on a JEM1400 transmission EM (JEOL) at 120 kV.

## 3. Results

### 3.1. Isolation of a Novel Paramyxovirus Isolate Using Primary Bat Kidney Cells

Previously, we described the isolation of an adenovirus and two previously unknown pararubulaviruses (AchPV1 and AchPV2) on Vero cells from a 72-strong sample batch of *Eidolon helvum* urine samples [5,11]. Here, we report an additional pararubulavirus isolated from urine sample number 72 (U72) of the same field collection batch using primary bat kidney (PaKi) cells. The pooled urine sample U72 had previously been negative for the *Paramyxovirinae* consensus PCR but positive for the consensus PCR for respiro-, morbilli-, and henipaviruses [15], and did not give rise to a paramyxovirus isolate on Vero cells [11].

However, on the fifth day post-infection of the second passage, a subtle CPE comprising syncytia formation and multinucleate PaKi cells was observed for sample U72 (Figure 1). Subsequent RT-PCR on RNA extracted from the supernatant of this flask (in which a novel previously undetected virus had amplified) was now positive by *Paramyxovirinae* consensus PCR and negative for the consensus PCR for respiro-, morbilli-, and henipaviruses [11,15]. Sequencing and phylogenetic analysis of the PCR fragment revealed the virus to be a novel pararubulavirus (Figure 2), and the virus was called Achimota pararubulavirus 3 (AchPV3) after the local area in which the samples were collected and to align it with other paramyxoviruses isolated from these urine samples [11].

### 3.2. AchPV3 Grows on Both Primary Bat Kidney and Vero Cells

In common with AchPV1 and AchPV2, AchPV3 grows on both primary bat cells and Vero cells [11]. Initially, AchPV3 was isolated from sample U72 on PaKi cells but not on the parallel isolation attempt on a Vero cell monolayer. The first passage PaKi cell monolayer was frozen and thawed before attempting to propagate further on both PaKi and Vero cell monolayers (methods, [11]). On this second passage, PaKi cells developed a subtle CPE typified by syncytia and multinucleate cell formation on day three post-infection. Since no CPE was observed in the parallel Vero cell monolayers, a working viral stock was harvested from the PaKi cells only (on day six post infection by purifying dilution, as previously described [11]). When generating a viral stock for full genome sequencing, however, PaKi cells were used and infection of Vero-E6 cells was also attempted. During this resuscitation, flasks were checked daily and syncytial CPE was observed in the PaKi cells on day 6 post-infection and in the Vero-E6 cell monolayers at 12 days post-infection.

### 3.3. Genomic Organisation

To characterise the AchPV3 genome and to explore the possibility of genomic signatures of the cell line adaptation, the whole genomes of AchPV3 grown on both Vero-E6 and PaKi cells were sequenced, as previously described for AchPV1 and AchPV2 [11]. These viral genomes derived from different cell lines were identical, and the AchPV3 genome sequence was deposited in Genbank under accession number: MT062420.

AchPV3 has a similar genomic organisation to existing pararubulaviruses. The genome is 15,600 nucleotides (nt) in length, making it fractionally shorter than AchPV1 (15,624) and slightly longer than AchPV2 (15,504). Overall, however, it had a similar genome structure to AchPV1 and AchPV2, encoding eight proteins: the nucleocapsid (N, 513 aa in length), matrix (M, 376 aa), fusion (F, 527 aa), attachment (HN, 587 aa), and polymerase (L, 2273 aa) proteins, as well as the overlapping V protein (238 aa), W protein (168 aa), and phosphoprotein (P, 395 aa) (Figure 3). These overlapping reading frames are facilitated by the existence of an RNA editing site (TTTAAGAGGGG) at position 2410 of the genome with the addition of none, one, and two non-templated guanine residues encoding the V, W, and P proteins, respectively.

AchPV3 also had a highly similar 3′ leader sequence to other para- and ortho-rubulaviruses, which was the reverse complement of its 5′ trailer sequence for the first 15 nt (Figure 4). Similar to AchPV1 and AchPV2, AchPV3 had a conserved transcriptional start signal (typically with a guanine residue in the +1 position in common with other pararubulaviruses), a conserved transcriptional stop signal, and intergenic regions (IGR) of variable lengths (Table 2).

### 3.4. Relationship with Other Paramyxoviruses

AchPV3 is related to, but distinct from, previously described pararubulaviruses. Phylogenetic analysis of the full-length N protein of AchPV3 demonstrated that the virus clustered phylogenetically with AchPV2, and further expanded the pararubulavirus genus (Figure 5). Phylogenetic comparison of the available partial polymerase gene sequences also showed AchPV3 to be distinct from the viral sequence fragments previously detected using consensus PCRs from different bat populations, from the same bat population, and from the same urine sample [15,18,19,20] (Figure 2). Amino acid sequence identities of the AchPV3 proteins were highest with other pararubulaviruses, followed by other rubulaviruses when compared with members from other paramyxoviral genera (Table 3). It was not confirmed whether the attachment protein had hemagglutinin or neuraminidase activity. However, phylogenetic analysis revealed that the AchPV3 attachment protein is most closely related to the AchPV2 attachment protein and clustered with those of other pararubulaviruses (Figure 6). Protein sequence analysis revealed that the AchPV3 attachment protein conserved the rubulavirus hexapeptide sequence (found at amino acid position 214 in the AchPV3 attachment protein) but does not share the conserved NRKS motif that is thought to be required for neuraminidase activity [27].

### 3.5. Electron Microscopy

EM analyses of PaKi cell cultures infected with AchPV3 confirmed the presence of a virus belonging to the family *Paramyxoviridae.* Examination of the cells in thin section revealed non membrane bound, cytoplasmic inclusion bodies homogenously packed with intertwined ribonucleic protein (RNP) (Figure 1). In a transverse section, individual RNP segments can be seen as hollow tubes (Figure 1D). The inclusion bodies, together with the observation of pleomorphic particles containing multiple strands of characteristic herringbone RNP in negative contrast EM, support the identification in the family *Paramyxoviridae*. In the negative contrast EM, the outer envelope appeared refractile to stain; consequently, the nature of the envelope was unable to be described. The exit strategy of the virus from the cell was also unable to be elucidated by EM at this stage of infection.

## 4. Discussion

Here, we have described a novel paramyxovirus from African fruit bats, AchPV3. We have sequenced and annotated its genome, visualised its structure, as well as characterised its evolutionary relationships with relevant *Paramyxoviridae*.

Phylogenetic and genetic analysis confirmed that AchPV3 is a member of the largely bat-derived *Pararubulavirus* genus, which is known to include zoonotic pathogens, suggesting that AchPV3 is a novel potential zoonotic virus. At 15,600 bp in length, the genome is exactly divisible by six (a known constraint of paramyxoviral genome architecture) and AchPV3 was found to use a conserved mRNA editing site to create three coding sequences within the P gene [28]. AchPV3 also exhibited paramyxoviral features under light and electron microscopy, including the CPE of syncytial cell formation, cytoplasmic inclusion bodies, and RNP.

Phylogenetic analysis of AchPV3 alongside other paramyxoviral polymerase protein sequences [15,18,19,20] demonstrates the existence of many other pararubulaviruses in this field sample collection batch, and throughout the Old World (Figure 2). Interestingly, the sample from which AchPV3 was isolated (U72) was initially positive for the respiro-, morbilli-, and henipa consensus PCR and negative for the *Paramyxovirinae* consensus PCR [15]. However, during virus culture, a pararubulavirus that amplified with *Paramyxovirinae* consensus PCR was isolated and no respiro-, morbilli-, or henipaviruses were isolated. The most likely explanation for this is that the *Paramyxovirinae* consensus PCR had insufficient sensitivity to detect AchPV3 in the original urine sample (the *Paramyxovirinae* subfamily consensus PCR is known to have 10–50 fold poorer sensitivity than the genus-based consensus PCRs like the respiro-, morbilli-, and henipa consensus PCR [24]) but that, once propagated in cell culture, the *Paramyxovirinae* PCR could detect AchPV3. The lack of isolation of the originally detected respiro-, morbilli-, henipavirus from sample U72 was a common feature across this body of work with no respiro-, morbilli-, or henipaviruses being isolated from attempted isolations from 19 urine samples that were PCR positive for viruses belonging to these genera, although attempts were only made using two cell lines [11,15]. Ultimately, the lack of molecular detection of a subsequently isolated virus (AchPV3) is further evidence that our current understanding of the diversity of pararubulaviruses (being several-fold higher than the number of live isolates) is likely still an underestimate.

Given the wide distribution and diversity of pararubulaviruses and the zoonotic nature of some members of this genus, studying the determinants of host restriction in this genus is a crucial avenue of research. To that end, we determined that AchPV3 was able to infect both PaKi cells and E6Vero-E6 cells, in common with its nearest phylogenetic relatives AChPV1 and AchPV2 [11]. Genomically identical viral isolates were recovered from both cell lines. Further, to determine the potential host range of AchPV3, the phylogenetic clustering of the AchPV3 attachment protein with those of other pararubulaviruses, along with the lack of conservation of a key hexapeptide motif, indicate that AchPV3 is likely to use a sialic-acid independent mode of cell entry in common with other pararubulaviruses [29]. Furthermore, the AchPV3 attachment protein clusters most closely with the AchPV2 attachment protein, which might indicate a common mode of entry and host range for these two viruses. This is of concern as AchPV2, distinct from AchPV1, appears to be capable of cross species infection infecting humans [11] and rodent and mustelid laboratory animal species [12]. The commonalities between the AchPV3 attachment protein and those of other pararubulaviruses, particularly AchPV2, is further evidence supporting a novel entry mechanism for pararubulaviruses that needs to be characterised if we are to better understand the host range of this emerging viral genus.

Here we have reported the isolation of a novel pararubulavirus, contextualised it within the samples from which it was isolated, and characterised its genome, its evolutionary relationships within the *Paramyxoviridae*, and its receptor-binding protein. Genome sequencing and analysis as well as EM revealed features typical of the genus. The lack of detection of the virus in the original sample and the close attachment protein relationship with a virus capable of multi-species, including zoonotic, infection adds to the complex picture of pararubulaviruses being a highly diverse, under detected, and widely distributed bat-derived viral lineage containing members of variable zoonotic potential. As such, the continued study of AchPV3 and other pararubulaviruses is an important area of research that will enhance our understanding of the determinants of host restriction and, ultimately, viral spillover from bats.

## Figures and Tables

**Figure 1 viruses-12-01236-f001:**
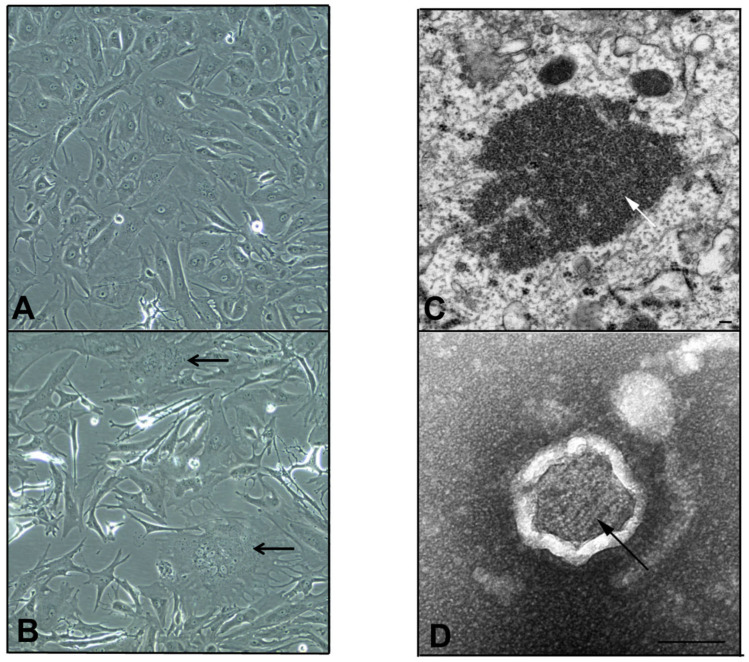
Growth of AchPV3 in culture by light (**A**,**B**) and electron (**C**,**D**) microscopy. *Pteropus alecto* primary kidney cells uninfected (**A**) and infected with AchPV3 (**B**), resulting in multinucleate syncytial cells (black arrows, **B**). Transmission electron micrographs (**C**,**D**) of thin sections reveal cytoplasmic inclusions of viral RNP (white arrow, **C**). In the negative contrast analysis, pleomorphic particles containing the characteristic *Paramyxovirus* ribonucleoprotein (RNP) were observed (black arrow, **D**). Scale bars represent 100 nm.

**Figure 2 viruses-12-01236-f002:**
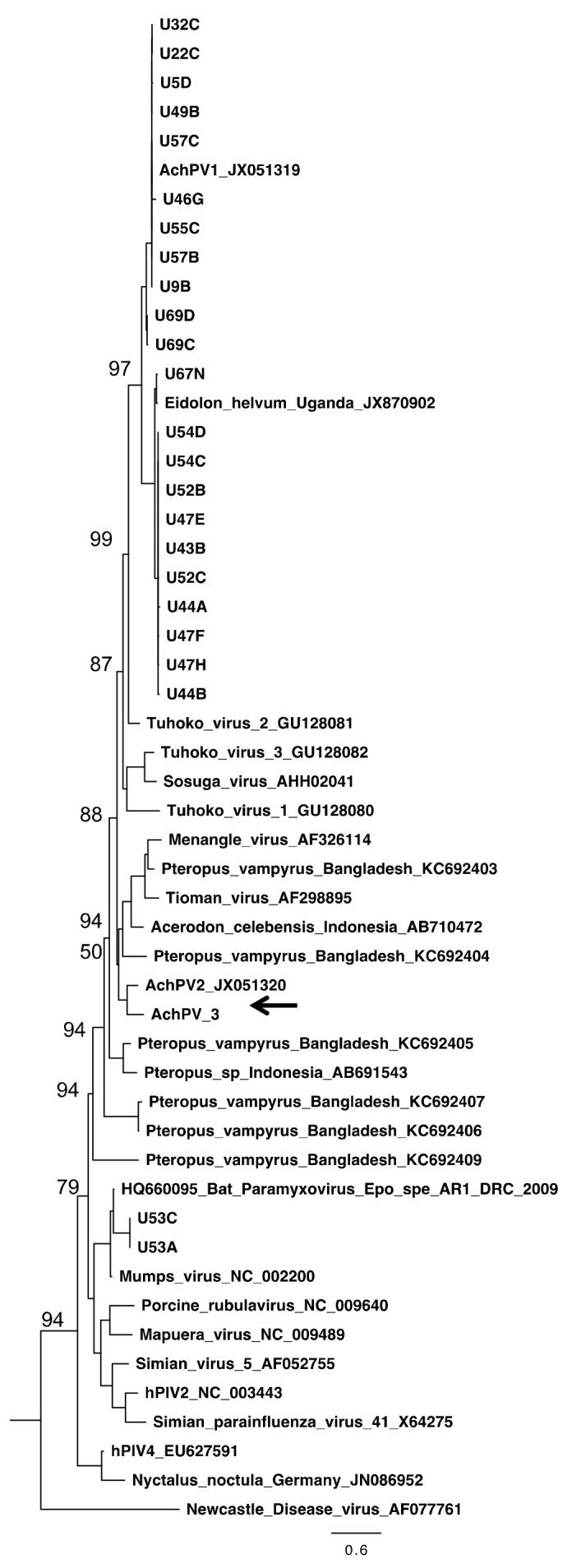
The phylogenetic relationship of AchPV3 polymerase protein and other paramyxoviruses detected among urine samples using consensus *Paramyxovirinae* PCR. The phylogenetic tree shows a 176 amino acid alignment of a polymerase protein fragment, and rubulavirus and pararubulavirus fragments detected from bats. Samples starting with U are from *Eidolon helvum* in Ghana [15]. The AchPV3 sequence is indicated by a black arrow and the scale bars represent the expected number of substitutions per site. Bootstrap values (of 100) of the relevant sites are shown. AchPV1 was detected in, and isolated from, sample U46; AChPV2 was detected in, and isolated from, sample U69; and AchPV3 was isolated from sample U72 but was not detected during PCR screening of the original sample.

**Figure 3 viruses-12-01236-f003:**
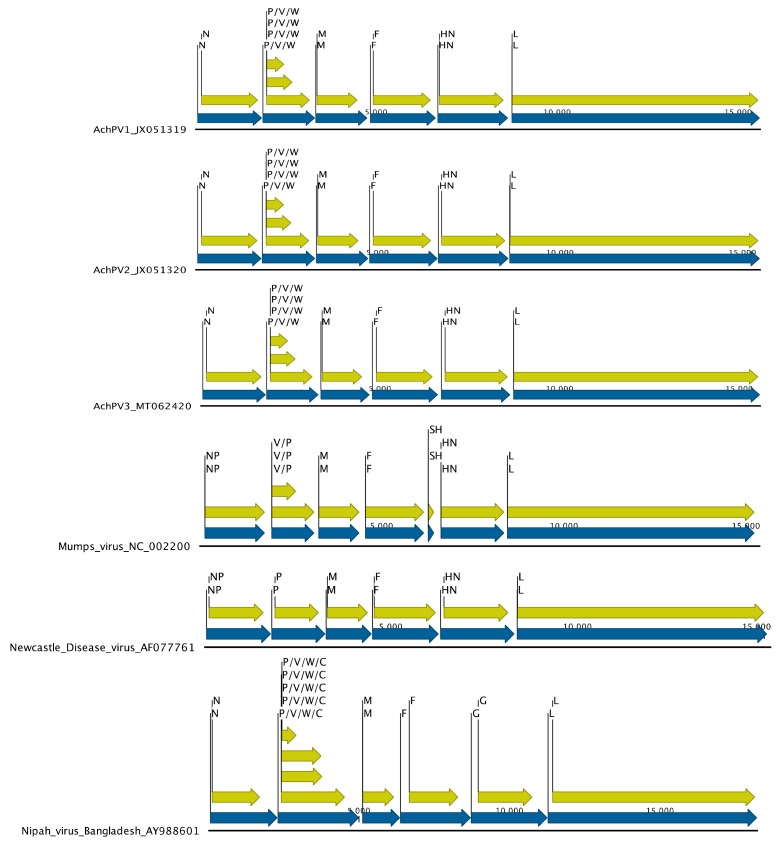
Genome architecture of Achimota viruses 1, 2, and 3, Mumps virus, Newcastle Disease Virus, and Nipah virus. Gene boundaries and orientations are indicated by blue arrows and coding sequences by yellow for the nucleoproteins (N/NP), phosphoproteins (P), V/W/C proteins, matrix proteins (M), fusion proteins (F), small hydrophobic protein (SH), attachment proteins (HN/G), and polymerase proteins (L).

**Figure 4 viruses-12-01236-f004:**
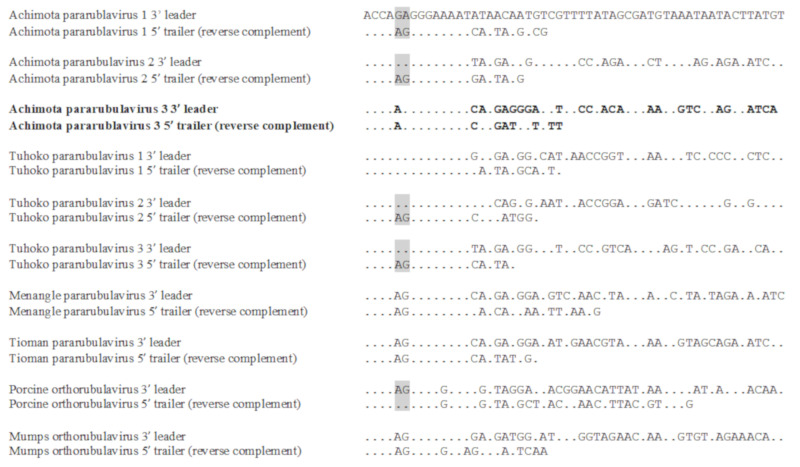
The leader and trailer sequences of various pararubulaviruses and the orthorubulaviruses, Porcine orthorubulavirus, and Mumps orthorubulavirus. Those places marked with a dot are where the sequence is identical to Achimota pararubulavirus 1. For viruses where the reverse complementarity of the 5′ trailer (to the 3′ leader) is compromised by an AG couplet, this is indicated by a grey box. AchPV3 is indicated in bold.

**Figure 5 viruses-12-01236-f005:**
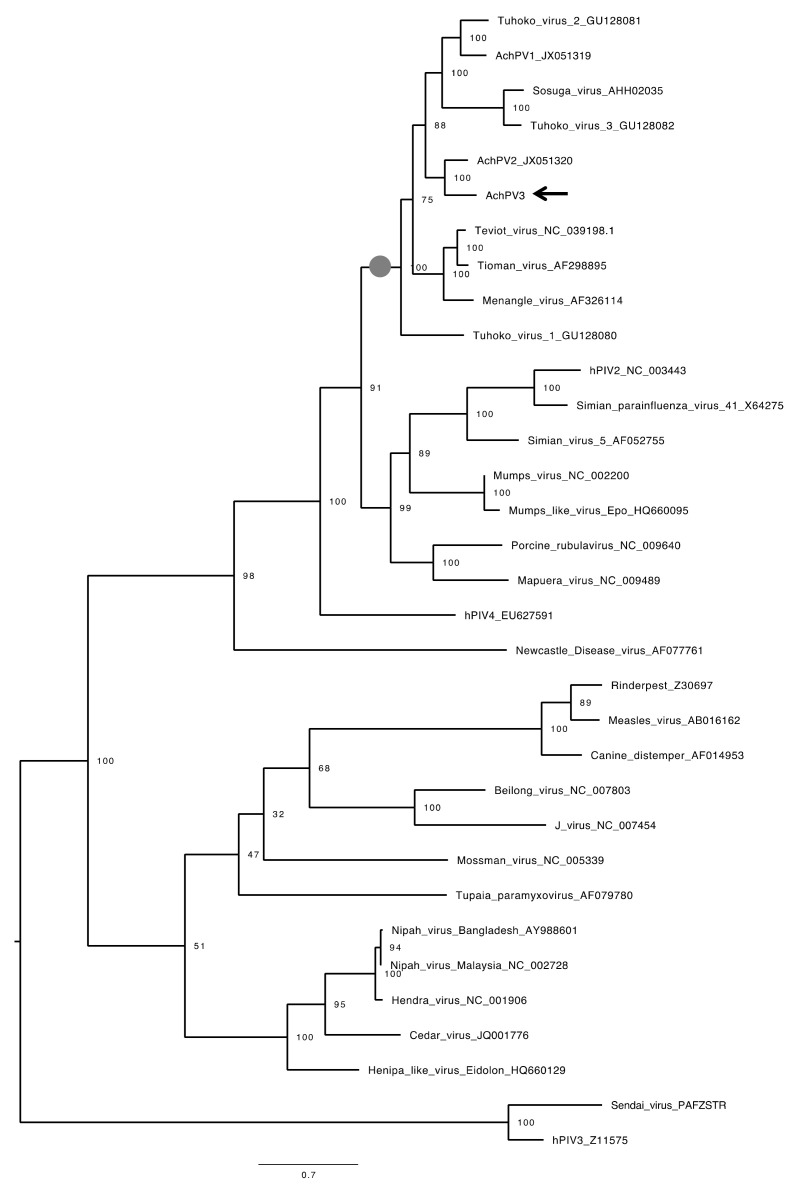
The phylogenetic relationships of AchPV3 among the paramyxoviruses. Midpoint-rooted maximum likelihood phylogenetic trees based on a 585 amino acid alignment of the nucleoprotein of *Paramyxovirinae* members where the pararubulavirus genus is demarcated by a grey circle on the internal node. The AchPV3 sequence is indicated by a black arrow and scale bars represent the expected number of substitutions per site, and the bootstrap values (of 100) of the relevant sites are shown.

**Figure 6 viruses-12-01236-f006:**
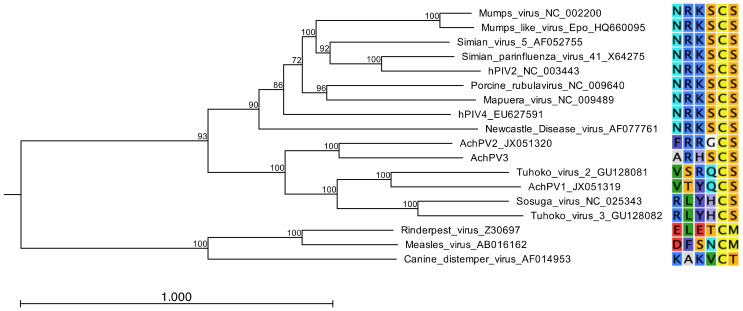
The phylogenetic relationship of the AchPV3 Hemagglutinin protein among other paramyxoviruses (pararublaviruses, orthorubulaviruses, and morbilliviruses). Midpoint-rooted maximum likelihood phylogenetic tree based on a 581 amino acid alignment and the scale bar is in expected substitutions per site. Adjacent to the label names is the hexapeptide motif found in the paramyxoviral attachment proteins.

**Table 1 viruses-12-01236-t001:** Primer sequences used in this study (from [24])**.**

Target	Primer	Sequence
*Paramyxovirinae*	Forward 1	GAAGGITATTGTCAIAARNTNTGGAC
	Forward 2	GTTGCTTCAATGGTTCARGGNGAY AA
	Reverse	GCTGAAGTTACIGGITCICCDATRTTNC
*Respirovirus Morbilliovirus Henipavirus*	Forward 1	TCITTCTTTAGAACITTYGGNCAYCC
	Forward 2	GCCATATTTTGTGGAATAATHATHAAYGG
	Reverse	CTCATTTTGTAIGTCATYTTNGCRAA

**Table 2 viruses-12-01236-t002:** Gene boundary sequence motifs and IGR features for the AchPV3 genes.

Gene	Start	Stop	IGR Sequence Boundaries	IGR Length (nt)
Consensus ^a^	gGGCCcGA	tTTTAAgAAAAAA		
N	AGGCCCGA	TTTTAAGAAAAAA	GGGAAAATAAAGGT	14
P	GGGCCCGA	TTTAAGAAAAAA	TGTGAAC...TGGAAGACT	54
M	GGGCCCGA	TTTTAAGAAAAAA	TATGCCC...CACCATAGT	63
F	GGGCCCGA	TTTTAATAAAAAA	CTGAGTTCT...AATAAAGGT	82
HN	GGGCCCGA	TTTAAGAAAAAA	GTGACATTA...TGTCATACT	75
L	GGGCCAGA	TTTAAGAAAAAA	TATCTAG...5′ trailer	21

^a^ Uppercase letters denote the base conserved across all genes; lowercase letters show the majority base where variable.

**Table 3 viruses-12-01236-t003:** Pair-wise amino acid identities for AchPV3 nucleocapsid (N) and phosphoproteins (P) with other Paramyxovirinae.

		Achimota Pararubulavirus 3	
Genus		N	P
*Pararubulavirus*	Achimota virus 1	66	43
	Achimota virus 2	75	47
	Tuhoko virus 1	60	38
	Tuhoko virus 2	66	39
	Tuhoko virus 3	60	41
	Menangle virus	64	41
	Tioman virus	65	39
	Sosuga virus	58	42
	Teviot virus	77	29
*Orthorubulavirus*	Mumps virus	51	24
	Mapuera virus	47	26
	Simian virus 41	45	25
	Human parainfluenzavirus 2	44	25
	Parainfluenzavirus 5	47	25
	Porcine rubulavirus	49	25
	Human parainfluenzavirus 4	42	24
*Morbillivirus*	Rinderpest virus	22	9
	Measles virus	23	8
	Canine distemper virus	23	10
*Henipavirus*	Hendra virus	27	7
	Nipah virus (Malaysia)	27	7
	Nipah virus (Bangladesh)	27	8
*Avulavirus*	Newcastle disease virus	31	20
*Respirovirus*	Sendai virus	19	5
*Jeilongvirus*	Beilong virus	25	10
	J-virus	22	9
*Narmovirus*	Mossman virus	25	8
	Tupaia paramyxovirus	23	8

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
