# Peer review of "Achimota Pararubulavirus 3: A New Bat-Derived Paramyxovirus of the Genus Pararubulavirus"

_viruses, 2020, doi:10.3390/v12111236_

Round 1
Reviewer 1 Report
Bats represent a major source of high variability in viruses with the potential to infect humans. This manuscript addresses the importance of understanding the genomic sequence and structure of paramyxoviruses harbored in bat species. The focus on rubulaviruses is potentially very significant as this paramyxovirus group has the potential for widespread infections of a diverse range of animal species.
Strengths of the work include the importance of determining viral sequences and types harbored in bats, the molecular sequence analysis of AchPV3 and comparison to other paramyxoviruses which show only ~60% identity with AchPV1 and AchPV2, and the clear microscopic evidence that there are nucleocapsid-associated virus particles in the preps.
The authors conclude that the AchPV3 is similar to other pararubulaviruses in genome structure. They should include a schematic of the structures of members of the paramyxovirus family so that the uninitiated can understand the comparative differences.
The main weakness of the work is the lack of depth in many of the reported results. For example, the authors appear to be surprised in the what they see as differences between time-to-appearance of syncytia between the bat cells and the Vero cells. This could be due to a number of factors. This reviewer does not believe there is enough data to conclude as is stated in the abstract “the growth of AchPV3 across different cell lines highlighted discrepancies in infection.” A similar claim in the discussion (lines 210) is not well justified. The authors did not measure infection or growth, but rather looked for CPE which was neither quantified or shown in data. For example, because the authors have RT-PCR in hand, the growth of this virus in terms of RNA synthesis (genomes, mRNAs) could be examined in different cell types.
The lack of depth in the reported results is also seen in the analysis of amino acid sequences in Table 1. What about the cis-acting genomic sequences such as leader and trailer promoters, intergenic regions, etc? Is the F protein cleavage site containing multiple basic residues? Are there differences in the attachment protein which would suggest a different receptor? These are a few of the examples of increasing the depth of analysis on this beyond simply comparing amino acid identity.
In short, while there are a number of interesting points in the manuscript, there is a lack of depth in this very short and superficial report.
Other Items:
- The materials and methods do not note whether the lab work was under BSL2 or BSL3 conditions.
- It is not clear from the materials and methods if the virus and cells used for sequencing is clonal or is a mixture of viruses. It is unclear whether there was any purification of virus (e.g., plaque?). This needs clarity as the authors state that the U72 was positive for respire, morbilli, henipa (line 129). This contrasts with line 133 which says the sample was negative for these concensus sequences.
- In Fig. 1B, the multi-nuclear syncytia are not clear. Can this be shown at higher magnification?
Minor Weakness:
- Line 33: Unknown spelling
- Line 62: demonstrates spelling
- Line 152: These viral genomes derived from different cell lines
- Line 164-166, clarify
- Line 186: novel candidate for potential
Reviewer 2 Report
Baker et al. described the isolation of Achimota pararubulavirus 3 from an African straw coloured fruit bat. The authors provided information on the sequence and genome organization of this novel virus and the phylogenetic relatedness to ortho- and pararubulaviruses.
Major Points:
- Provide the primer sequences used for consensus PCR in the methods part.
- When focusing on HN gene, pararubulaviruses cluster outside the group of orthorubulaviruses. It is further assumed that particular pararubulaviruses do not interact with sialic acids, conclusively pararubulaviruses might express attachment glycoproteins that are different from the HN proteins found in orthorubulaviruses. Include additional data showing the phylogenetic relatedness of the AchPV3 attachment protein to address the question which kind of attachment glycoprotein might be present for this novel virus.
Minor Points:
- please correct typing errors: line 33 "unknonw", line 62 "ddemonstrate", line 126 "paraubulavirus"
Reviewer 3 Report
Baker et al. discovered and isolated a new paramyxovirus, Achimota pararubulavirus 3 (AchPV3), from the urine of African straw coloured fruit bats using a bat primary cell , PaKi cells. Based on the whole genome analyses, the authors found that the AchPV3 is classified as a new member of the genus, Pararubulavirus. Since the other members in the genus are derived from bats and known to cause disease in human, the authors speculated that the isolated virus is one of the causative pathogens for zoonosis. Then, the authors focused and described a discrepancy in growth of the virus, based on the observation that the AchPV3, which was originally isolated on the PaKi cells, not the Vero cells, could grew in Vero cells. Basically, the virus discovery is of interest, while the methodology of molecular analyses in the study are adequate and convincing to support the result. However, information provided by the discovery might be limited, because the authors had already reported the close relatives of the virus, the AchPV1 and AchPV2, from the same pool of samples. Moreover, the structure of the manuscript, especially for the descriptions on the above discrepancy in infection, should be re-considered. The descriptions are quite confusing, and I do not consider that the authors provided enough data to discuss features of the virus in growth.
Specific points to be considered are as follows:
- Line 34: To better prepare for “these” outbreaks
It would not necessarily be vital to discovering a new bat-derived virus (e.g. a novel virus in Paramyxovirus) for preparedness to the specific viruses (Ebola virus etc).
- Line39: Reference [2,3,4]
Replace with more appropriate references describing henipaviruses.
- Line 62: “ddemonstrates”
Correct the word.
- Line 93: media was topped-up
The medium was replaced with the fresh medium? Or simply the medium was added to the virus inoculum? High MOI might influence virus growth.
- Line 100: “CSIRO AAHL illumina Miniseq sequencing system”
This is the specific name for the system?
Or this means that the method or protocol is established in the facility?
- Line145: “When generating a viral stock for full genome sequencing”
The virus stock was harvested on the third passage? Clarify this.
- Line146: “Surprisingly, syncytial CPE,,,,”
What is the surprise? Does the sentence mean that the syncytia observed on the cells disappeared after 6 dpi, due to detachment of the fused cells? Explain in more detail.
- Line 152-153: “These viral genomes,,,,”
Sequence identity among the viruses grown on different cell lines showed that the AchPV3 could grow even in the Vero cells without any adaptation. Also, it would be suggested that the virus grows less on the Vero cells, compared with its growth on the PaKi cells, because the virus was only isolated from the PaKi cells, not from the Vero cells. If the authors focus the unique feature of the virus in the manuscript, different from that of the AchPV1 or AchPV2 (as described in the line 199-211.), it seems rather easy to include data regarding to the growth kinetics (growth curves) on these cells.
- Line 149: 3.3 Genomic organization
It would be helpful for reader to understand the organization if the authors could provide an illustration or a Table.
- Figure 1:
Images A and B were obtained as Phase contrast? If so, clarify this in the legend, as well s in the line 189. The panels A and B, should be placed separately from the panels C and D. Also, scale bars or magnification might need to be shown for the panels A and B.
- Figure 2:
The phylogenetic tree for pol gene should be shown separately from that for N gene. Also, it should be shown in the legend that the sequence was determined sing the sample from U72 etc.
Round 2
Reviewer 1 Report
The authors have been very responsive to the prior critique and have carried out the following important modifications:
1) soften the conclusions on growth of viruses in different cell lines
2) increased the analysis of sequences:
An analysis of the leader and trailer sequences of AchPV3 among other
paramyxoviruses (Figure 4, Ls 201 - 202)
• A phylogenetic and proteomic analysis of the AchPV3 attachment protein, including detailed discussion of the implications of these findings (Figure 6, Ls 216 – 221, 312 –321)
• Analysis and annotation of conserved transcriptional start and stop sites and
intergenic regions of AchPV3 (Table 3, Ls 202 – 205)
Author Response
we thank the reviewer for this positive response to our revision
Reviewer 2 Report
The authors have addressed all of my comments.
Author Response

(The authors gave the same response as above.)

Reviewer 3 Report
The authors have provided adequate and full answers to all my comments/suggestions.
However, some points should be confirmed or modified before publishing.
- Figure 3 legend:
Provide information for the yellow and blue arrows. And explain the meaning of “IN” (iN, iM) etc.
- Figure 4:
It would be better to be spelled out for the names of virus (ThkPV, MenPV, TioPV, PorPV, MuV), as they are done in other Figures and Tables. Also “rev comp” is to be clarified such as “reverse complement”.
- Figure 4 legend:
Explain about the “.” (identical or deletion etc).
4.Table 2:
Clarify the meaning of “size(nt)”. Size of IGR sequence?
Explanation should be provided in the legend or in the manuscript.
5.Table 4:
Spell out the names of viruses (or explain the abbreviations).
Author Response
However, some points should be confirmed or modified before publishing.
- Figure 3 legend:
Provide information for the yellow and blue arrows. And explain the meaning of “IN” (iN, iM) etc.
This has been done via the addition of the following text to the figure legend “Gene boundaries and orientations are indicated by blue arrows and coding sequences by yellow for the nucleoproteins (N/NP), phosphoproteins (P), V/W/C proteins, matrix proteins (M), fusion proteins (F), small hydrophobic protein (SH), attachment proteins (HN/G), and polymerase proteins (L).”
- Figure 4:
It would be better to be spelled out for the names of virus (ThkPV, MenPV, TioPV, PorPV, MuV), as they are done in other Figures and Tables. Also “rev comp” is to be clarified such as “reverse complement”.
The names of the viruses and the abbreviation of ‘rev comp’ have now been fully expanded in the figure.
- Figure 4 legend:
Explain about the “.” (identical or deletion etc).
The figure legend has been updated to include this information. It now reads: “Figure 4. The leader and trailer sequences of various pararubulaviruses and the orthorubulaviruses, Porcine orthorubulavirus and Mumps orthorubulavirus. Those places marked with a dot are where the sequence is identical to Achimota pararubulavirus 1. For viruses where the reverse complementarity of the 5’ trailer (to the 3’ leader) is compromised by an AG couplet, this is indicated by a gray box.”
4.Table 2:
Clarify the meaning of “size(nt)”. Size of IGR sequence?
Explanation should be provided in the legend or in the manuscript.
We thank the reviewer for pointing out this confusing oversight. The text in the table has been changed to “IGR length (nt)” at the abbreviation for intergenic region (IGR) has been added to the text at first usage (L256). We have also added the abbreviation for nucleotides (nt) at first use at L244.
5.Table 4:
Spell out the names of viruses (or explain the abbreviations).
The viral names have been expanded in full in the table.